# CKAP2L Knockdown Exerts Antitumor Effects by Increasing miR-4496 in Glioblastoma Cell Lines

**DOI:** 10.3390/ijms22010197

**Published:** 2020-12-27

**Authors:** Yao-Feng Li, Wen-Chiuan Tsai, Chung-Hsing Chou, Li-Chun Huang, Shih-Ming Huang, Dueng-Yuan Hueng, Chia-Kuang Tsai

**Affiliations:** 1Department of Pathology, Tri-Service General Hospital, National Defense Medical Center, Taipei 11490, Taiwan; liyaofeng@ndmctsgh.edu.tw (Y.-F.L.); ab95057@hotmail.com (W.-C.T.); 2Department of Neurology, Tri-Service General Hospital, National Defense Medical Center, Taipei 11490, Taiwan; choutpe@yahoo.com.tw; 3Graduate Institute of Medical Sciences, National Defense Medical Center, Taipei 11490, Taiwan; shihming7102@gmail.com (S.-M.H.); hondy2195@gmail.com (D.-Y.H.); 4Department of Biochemistry, National Defense Medical Center, Taipei 11490, Taiwan; emily7781@hotmail.com; 5Department of Neurological Surgery, Tri-Service General Hospital, National Defense Medical Center, Taipei 11490, Taiwan

**Keywords:** *CKAP2L*, glioma, glioblastoma, prognosis, UCSC Xena, TCGA, CGGA, miR-4496

## Abstract

Despite advances in the diagnosis and treatment of the central nervous system malignancy glioma, overall survival remains poor. Cytoskeleton-associated protein 2-like (*CKAP2L*), which plays key roles in neural progenitor cell division, has also been linked to poor prognosis in lung cancer. In the present study, we investigated the role of *CKAP2L* in glioma. From bioinformatics analyses of datasets from The Cancer Gene Atlas and the Chinese Glioma Genome Atlas, we found that *CKAP2L* expression correlates with tumor grade and overall survival. Gene set enrichment analysis (GSEA) showed that MITOTIC_SPINDLE, G2M_CHECKPOINT, and E2F_TARGETS are crucially enriched phenotypes associated with high *CKAP2L* expression. Using U87MG, U118MG, and LNZ308 human glioma cells, we confirmed that *CKAP2L* knockdown with si*CKAP2L* inhibits glioma cell proliferation, migration, invasion, and epithelial-mesenchymal transition. Interestingly, *CKAP2L* knockdown also induced cell cycle arrest at G2/M phase, which is consistent with the GSEA finding. Finally, we observed that *CKAP2L* knockdown led to significant increases in miR-4496. Treating cells with exogenous miR-4496 mimicked the effect of *CKAP2L* knockdown, and the effects of *CKAP2L* knockdown could be suppressed by miR-4496 inhibition. These findings suggest that *CKAP2L* is a vital regulator of miR-4496 activity and that *CKAP2L* is a potentially useful prognostic marker in glioma.

## 1. Introduction

Central nervous system (CNS) malignancies are responsible for significant morbidity and mortality worldwide [1]. Among CNS malignancies, glioma is the most common, accounting for 26% of all CNS neoplasms [2]. The most lethal of the gliomas is the glioblastoma (GBM) World Health Organization (WHO) grade IV [3]. Before 2016, glioma diagnoses were based on cellular morphology and included astrocytoma, oligodendroglioma, and ependymoma, based on the 4th version of WHO Classification for Brain Tumors [4]. Since the revision of the 4th edition of the WHO Classification [3] and recent updates [5,6,7], the molecular information has played a critical role in prognosis prediction and treatment, such as *IDH1/2* [8,9], 1p19q [3], *EGFR* amplification [6], combined whole chromosome 7 gain and 10 loss [6], TERT promoter mutation [6], *CDKN2A/B* homozygous deletion [5], and *MGMT* methylation status [10]. For instance, methylation profiles were recently used to achieve better brain tumor classification for risk stratification [11,12]. However, despite these advances in molecular stratification, overall survival among patients with GBM remains poor at 14.6 months [13,14]. Hence, there is an essential need to identify new biomarkers and treatments that improve patient outcomes.

One group of potential targets is microRNAs (miRNAs), which are non-coding, single-stranded RNAs that contain around 21–25 nucleotides and act post-transcriptionally to influence gene expression [15,16,17,18]. In recent years, many significantly up/down-regulated miRNAs have been detected in GBM, including miR-21, miR10, and miR92 [16,18,19,20], to name just three. The actions of these miRNAs have been linked to cellular proliferation, survival, invasion, and drug resistance [16,17]. Targeting miRNAs is now considered to be a potentially effective way to modify the expression of oncogenes and tumor suppressor genes to stop or slow tumor progression.

Cytoskeleton-associated protein 2-like (*CKAP2L*) gene, located on chromosome 2, encodes a protein that is vital for cell division in neural progenitor cells [21] and is part of the centrosome, located in the spindle, midbody, and spindle pole [22]. Previous studies showed that loss of *CKAP2L* function leads to Filippi syndrome and microcephaly [23]. On the other hand, increased *CKAP2L* expression is associated with a poorer prognosis in pulmonary adenocarcinoma patients [24]. Up to now, the role of *CKAP2L* in gliomas has not been addressed. In the present study, therefore, we used bioinformatic analysis, clinical validation, cell modeling, and miRNA screening to investigate the role played by *CKAP2L* in gliomas.

## 2. Results

### 2.1. CKAP2L Expression Correlated with Tumor Grade and Overall Survival in Glioma

Statistical analysis showed a strong correlation between *CKAP2L* mRNA expression and prognosis in both TCGA and the CGGA datasets. Among the 701 TCGA samples, we found that *CKAP2L* expression correlated significantly (*p* < 0.0001) with tumor grade (Figure 1A). When the dataset was subdivided based on tumor grade (normal, Gr-II, Gr-III, and Gr-IV), *CKAP2L* mRNA expression significantly (*p* < 0.01) differed among groups, though no difference was detected between normal brain tissue (*n* = 5) and Grade II glioma (Figure 1A). This likely reflects the small number of samples of normal tissue. In addition, after dividing TCGA dataset into high-*CKAP2L* (>5.526, *n* = 346) and low *CKAP2L* (≤5.526, *n* = 346) subgroups based on the median *CKAP2L* expression, Kaplan–Meier analysis showed that higher *CKAP2L* mRNA expression was significantly (*p* < 0.0001) associated with a poorer prognosis (Figure 1B). Similarly, among the 325 glioma samples in the CGGA dataset, *CKAP2L* mRNA expression correlated significantly (*p* < 0.0001) with the tumor grade (Figure 1C). Moreover, after subdividing the dataset according to tumor grade (Gr-II, Gr-III, and Gr-IV), *CKAP2L* mRNA expression significantly differed among the groups (*p* < 0.0001). Furthermore, as with TCGA dataset, dividing the CGGA dataset into high-*CKAP2L* (>1.465, *n* = 162) and low-*CKAP2L* (<1.465, *n* = 163) expression groups based on the median expression showed that high *CKAP2L* expression was significantly (*p* < 0.0001) associated with a poor prognosis (Figure 1D). These results indicate that *CKAP2L* expression correlates with tumor grade and is associated with poorer overall survival. After further breakdown by the tumor grading (Appendix A), we found the grade III glioma revealed a survival difference between the high and low *CKAP2L* expression groups at TCGA and CGGA datasets (*p* < 0.001 and *p* = 0.017, respectively). In the grade II gliomas, the curves were separated, but only the TCGA dataset reached statistical significance (*p* < 0.001). In the GBM cluster, high and low *CKAP2L* expression groups’ curves only revealed borderline significance (*p* = 0.054).

### 2.2. MITOTIC_SPINDLE Is the Key Enriched Phenotype in the High CKAP2L Expression Group

To further understand the differences between the high and low *CKAP2L* expression groups, we performed gene set enrichment analysis (GSEA) with TCGA and the CGGA datasets. A total of 29 gene sets were significant at False Discovery Rate (FDR) < 25%, and 12 gene sets were significantly enriched at a nominal *p*-value < 5%. The top up-regulated phenotype in the high-*CKAP2L* group from TCGA (LogFC > 5.526, *n* = 351) was MITOTIC_SPINDLE, followed by DNA_REPAIR, G2M_CHECKPOINT, MTORC1_SIGNALING, and E2F_TARGETS (Figure 2A). Similar results were obtained with the CGGA dataset. The most up-regulated phenotypes in the high *CKAP2L* group among CGGA samples (LogFC > 1.465, *n* = 162) were E2F_TARGETS, G2M_CHECKPOINT, MITOTIC_SPINDLE, DNA_REPAIR, and MYC_TARGETS_V1. Thus, four of the top five phenotypes were the same in the two datasets (Figure 2B). The details of each result are demonstrated in the (Appendix A).

### 2.3. CKAP2L Staining Correlates with WHO Grade in Glioma Tissues

To evaluate CKAP2L protein expression in human glioma tissues, we immunostained tissue microarray slides containing gliomas of various WHO grades for CKAP2L. After eliminating cases that lacked adequate specimens, 93 samples were examined, including 10 samples of non-neoplastic brain tissue and 83 gliomas. IDH1, R132H, and H3K27M immunohistochemistry (IHC) staining showed that the glioma cases contained pilocytic astrocytoma, diffuse astrocytoma with/without IDH-mutant, anaplastic astrocytomas with/without IDH-mutant, oligodendrogliomas, anaplastic oligodendrogliomas, glioblastomas with/without IDH-mutant, and diffuse midline glioma H3K27M-mutant (Table 1). We observed that normal brain tissue samples showed negative or focally faint CKAP2L staining. By contrast, higher CKAP2L staining significantly correlated with tumor grade (*p* = 0.005, Figure 3A–E), further confirming that CKAP2L expression is associated with higher tumor grade. However, no correlation was found among specific subtypes (Table 1). To assess the relation between *CKAP2L* and glioma-related genes and other factors, we performed univariate and multivariate analyses. The results showed that age, *MGMT* methylation, *EGFRvIII*, *p53*, *NF1*, *AxL*, *p-AxL*, and *H3Lys27* expression were associated with *CKAP2L* upregulation (Table 2). In addition, Cox regression analysis revealed that older age, IDH1, loss of ATRX, and positive expression of *AxL*, *p-AxL*, *NUR77*, *H3Lys27,* or *PDGFRA* were independent prognostic factors (Table 3).

### 2.4. Increased CKAP2L Expression in Human Glioma Cells

To further explore *CKAP2L* mRNA and protein expression in glioma, we performed quantitative RT-PCR and Western blot analyses with normal brain tissue and the U87MG, U118MG, and LNZ308 human glioma cell lines. We found that *CKAP2L* mRNA expression was significantly higher in all three glioma cell lines than in normal brain tissue (*p* < 0.001, Figure 4A, and Appendix A). Likewise, expression of CKAP2L protein was higher in the three glioma cell lines than normal brain tissue (Figure 4B).

### 2.5. Suppression of CKAP2L Inhibits Glioma Cell Growth

Given that U87MG and U118MG glioma cells exhibit higher *CKAP2L* mRNA expression than normal brain tissue, we next tested whether glioma phenotypes (increased proliferation, invasion, and migration) could be suppressed by knocking down *CKAP2L* expression using small interfering RNA (siRNA). After transfection of si*CKAP2L* or siRNA (scramble siRNA) into U87MG and U118MG glioma cells, the expression of CKAP2L was substantially reduced at both 25 nM and 37.5 nM siRNA supplementation (Figure 5A,B). In order to avoid siRNA off-target effects, we chose 25 nM RNA for the following experiment. Evaluation of cell proliferation 24, 48, and 72 h after the siRNA transfection revealed that within 24 h in U87MG cells or 48 h in U118MG cells, cell number was significantly lower in siCKAP2L-expressing cells than in the control group (Figure 5C,D). To further investigate the relations between the cell decrease and proliferation ability, we looked at the association of the *CKAP2L* versus the commonly used proliferation biomarkers in the TCGA dataset, including *MKI67* [25] and *MCM6* [25,26,27] expression, which have been used in other brain tumor studies. The results demonstrated in Figure 6 showed the high CKAP2L expression positively correlates with both proliferation markers, MKI67 and MCM6 (R = 0.906 and 0.782; R^2^ = 0.82 and 0.661, respectively, and both *p* < 0.0001) in the TCGA dataset, as well as the CGGA dataset (R = 0.877 and 0.820; R^2^ = 0.769 and 0.672, individually; *p* < 0.0001). These findings supported the proposal that the CKAP2L expression is correlated with proliferation ability.

### 2.6. CKAP2L Knockdown Induces Cell Cycle G2/M Arrest

*CKAP2L* is known to be a component of the mitotic spindle [21], which helps to segregate the chromosomes during the mitosis phase (M phase). To investigate the impact of *CKAP2L* on mitotic function, we compared the cell cycle phase distributions between glioma cells expressing si*CKAP2L* and siControl. We found that among U87MG and U118MG cells, *CKAP2L* knockdown led to an increase in the numbers of cells in the G2/M phase and a dramatic decrease in cells in the G1 phase (Figure 7A,B). Consistent with that idea, Western blot analysis showed that cellular levels of cyclin B1 and p-cdc2 were elevated by *CKAP2L* knockdown in both U87MG and U118MG cells (Figure 7C–E). Because *CKAP2L* knockdown reduced cell proliferation, despite the increase in cells at the G2/M phase, we suggest that the glioma cells were arrested at the G2/M phase.

### 2.7. CKAP2L Knockdown Inhibits Invasiveness and Epithelial-Mesenchymal Transition (EMT) in Glioma

To detect the relationship between *CKAP2L* expression and glioma cell malignancy, cell invasion and wound-healing assays were carried out with U87MG glioma cells. In the invasion study, fewer si*CKAP2L*-expressing cells migrated through the transwell membranes, suggesting that *CKAP2L* suppression reduced the invasiveness of these tumor cells (*p* < 0.05, Figure 8A). Similarly, wound healing was slower in cultures of *CKAP2L* knockdown cells than control cells. Sixteen hours after wounding, the scratch area was significantly larger in cultures of *CKAP2L* knockdown cells (*p* < 0.001, Figure 8B). In addition, evaluation of three EMT biomarkers (E-cadherin, Twist, and Vimentin) revealed a significant decrease in Twist in *CKAP2L* knockdown cells (Figure 8C,D). This suggests a decrease in EMT after *CKAP2L* knockdown, which also could contribute to the observed reduction in invasiveness and slower migration.

### 2.8. CKAP2L Knockdown Decreases Cell Numbers by Increasing miR-4496

Using a miRNA array to analyze miRNA expression in U87MG and U118MG glioma cells expressing si*CKAP2L* or siControl, we identified three miRNAs that were upregulated after *CKAP2L* knockdown in both cell types: miR-4496, miR-642b-3p, and miR-4740 (Table 4). Among these three, the greatest increase was in miR-4496 levels, which were increased 4.469-fold (Log2FC = 2.16) in U87MG cells and 5.24 (Log2FC = 2.39) in U118MG cells (Table 4). To investigate the relation between miR-4496 and *CKAP2L*, we first tested the effect on cell proliferation of exogenously applying miR-4496 to the two cell lines. As shown in Figure 9A, cell numbers were significantly decreased when cells were exposed to 10 nM miR-4496 for more than 48 h. To further confirm that these cellular reductions were not caused by cell death, the cell cycle analysis was performed. The results showed that the sub-G1 phase was very low when the concentration of miR-4496 was 0, 20, or 30 nM (Figure 9B). However, an inconspicuous band showed when the contrition reached 40 nM, indicating that notable cell death had occurred. Consequently, we presumed that the decreased cell number at the 10 nM miR-4496 (Figure 9A) was not relevant to cell death but more likely to be associated with proliferation impairment. Thus, the effect of miR-4496 on glioma cell proliferation was similar to that of *CKAP2L* knockdown. This suggests that suppressing *CKAP2L* led to increases in miR-4496, which reduced glioma cell proliferation. To test that idea, we assessed the effect of the miR-4496 inhibitor. We found that in the presence of miR-4496 inhibitor, there was a dose-dependent decline in the difference between cells expressing si*CKAP2L* and those expressing siControl (Figure 9C). This suggests that suppressing *CKAP2L* decreased cell proliferation in part by increasing levels of miR-4496, and that effect was reversed by miRNA-4496 inhibition.

## 3. Discussion

At present, there is only limited information about the role of *CKAP2L* in glioma. Cytoskeleton-associated protein 2 (*CKAP2*) is an important paralog of *CKAP2L* that functions to stabilize microtubules and plays a key role in the regulation of cell division [28,29]. *CKAP2 gene* is also upregulated in various kinds of cancer, including breast [30], prostate [31,32], stomach [33,34], and ovarian [35] cancers, as well as glioma [36]. We therefore hypothesized that *CKAP2L* may play similar roles in glioma. Interestingly, *CKAP2L* also appears to function during mitotic phases and to play a critical role in cell division. Furthermore, an earlier study showed that downregulation of *CKAP2L* triggers multipolar spindles and cell cycle arrest [37], which is consistent with our present findings in glioma cells and tissues.

From TCGA and CGGA glioma datasets, we noticed the difference in CKAP2L expression showed high significance (*p* < 0.0001) among the tumor grades. However, the error bars (Figure 1A,C) are huge. One of the probable causes might be these grades were separated based on the 4th version of the WHO Classification for Brain Tumors [4]. However, each grade of the tumor is still heterogeneous. From the current point of view [3], the previous classification could be further stratified by integrating molecular information [5,6,7]. For example, the grade II glioma included oligodendroglioma, diffuse astrocytoma with/without IDH mutant [3], glioma with glioblastoma molecular features [5,6], and cases with better outcome [7]. Similarly, the grade III group is also heterogeneous and could be separated into different prognostic groups. Hence, each grade of the tumor is heterogeneous and mixed with good and bad prognosis cases, which might express different levels of CKAP2L and consequently, a large error bar. However, there is no useful molecular information to help break down both TCGA and CGGA datasets currently.

Furthermore, in the Kaplan–Meier plots, it seems that the CGGA has better overall survival in both low and high *CKAP2L* at the end. To make a better comparison between both datasets, we rescaled the Kaplan–Meier plots (Appendix A). From these plots, we did notice that the overall survival of both low and high *CKAP2L* seems to be better in the CGGA, although the proportion of grade IV in CGGA is higher than TCGA. However, if we checked overall survival time before the 4000 days, both trends were similar or slightly better at the TCGA dataset. Additionally, the median cut-offs were very different in both datasets. In the TCGA, the median *CKAP2L* expression was 5.526, but equal to 1.465 in the CGGA. To our knowledge, the data from CGGA used the same mRNA-seq pipeline as the TCGA database from a similar platform (Illumina HiSeq 2000/2500) to generate expression data [38] and both quantified in transcripts per million (TPM). Hence, the possible reasons for the different cut-off might be due to different case numbers in each grading. Additionally, another potential reason is that the level of sequencing coverage is unclear and this might cause the difference in the values. Not surprisingly, a similar issue was also noted in another glioma investigation [39]. Nevertheless, if we ignore the absolute value and only separate them by the median, their overall survival trend was similar (Appendix A).

In previous sections, we have introduced the proposal that *CKAP2L* plays a vital role in regulating cell division [37]. To further understand the impact of the *CKAP2L* expression on cell growth, we investigated the effect of *CKAP2L* knockdown on the proliferation in U87MG and U118MG glioma cells. From our data, we found *CKAP2L* knockdown suppresses U87MG and U118MG cell growth. To further support the proposal that the decrease of cell number was due to the decline of proliferation ability, we looked at the association of the *CKAP2L* versus two common proliferation biomarkers in the TCGA and CGGA datasets, including MKI67 and MCM6, often used in brain tumor studies [25,26,27]. The results demonstrated in Figure 6 showed that the *CKAP2L* expression positively correlated with *MKI67* and *MCM6* in the TCGA and CGGA. These findings support the proposal that the CKAP2L expression is associated with proliferation capability. Additionally, from the results of GSEA analyses, the E2F, mitotic spindle, and G2/M checkpoint are the leading phenotypes in the high CKAP2L group. To our knowledge, the E2F is the crucial regulator of the G1/S transition [40], and the G2/M checkpoint is essential for cell mitosis, as well as the mitotic spindle. These pieces of evidence all point out the changes in the proliferation ability when the CKAP2L is different. Hence, we presumed that a lower cell number in the siCKAP2L group than the control was due to decreased proliferation ability.

Next, to further model the consequence of the *CKAP2L* expression on the cell cycle, we examined the effect of *CKAP2L* knockdown on the cell cycle profile in U87MG and U118MG glioma cells. The results showed that the cell fraction in the G2/M phase was dramatically increased in both cell lines, while the fraction in the G1 phase was decreased (Figure 7A,B). We also examined the cell cycle’s regulated proteins for validation, including Cyclin D1, Cyclin B1, and p-cdc2 (CDK1). Our result showed the level of cyclin B1 and p-cdc2 protein underwent a significant increase in the si*CKAP2L* treatment group (Figure 7C). Typically, the shift from G2 to the M phase is triggered by the active cyclin B1/cdc2 complex. In other words, suppression of the function of cyclin B1/cdc2 causes cell arrest [41]. The increase of the cyclin B1/cdc2 complex was expected to trigger cell proliferation from our data. However, the cell growth investigation revealed much less cell proliferation after being supplanted with si*CKAP2L* compared to the control group (Figure 5C,D). The phenomenon was also noted in another breast cancer study [42]. In previous publications, the rapid and excessive increase of the cyclin B1-dependent/cdc2 in the G2 phase could cause cells to stop entering the mitotic phase [43,44]. One publication even provided evidence that the accumulation of cyclin B1 and cdc2 could induce chromosomal nuclear condensation, segregation, and then blocked cells in prometaphase [42]. Interestingly, from the results of GSEA analyses, the G2/M checkpoint’s alteration is a leading phenotype in cells expressing a high level of *CKAP2L*. On the other hand, the lower expression of *CKAP2L* might decrease the function of the G2/M checkpoint. Despite the increase in cells in the G2/M phase, cell proliferation was significantly reduced; we therefore hypothesized that the arrested cells were blocked at the G2/M checkpoint due to the possible presence of multipolar spindles [37] that may be caused by suppression of *CKAP2L* expression. Moreover, apart from the G2/M alteration in cell cycle analysis, we also noted that the G1 phase fraction decreased (Figure 7A,B), matching with the reduced function of E2F in the lower *CKAP2L* group identified in the GSEA analysis (Figure 2). Furthermore, in the cell cycle’s protein validation, we also noticed the variation of cyclin D1, which is another critical contributor in the G1/S transition. Compared to the U118MG, the U87MG supplemented with si*CKAP2L* showed a much lower cyclin D1. Consistently, the cell proliferation rate also showed more suppression in the U87MG (Figure 5C,D).

Our immunohistochemical staining showed that levels of *CKAP2L* protein correlated with the mRNA levels and that *CKAP2L* upregulation was associated with higher tumor grades, which was consistent with the bioinformatics analysis of TCGA and the CGGA datasets. In addition, our survival analysis suggested that higher *CKAP2L* levels were associated with a poorer prognosis. However, the significance level was weak (*p* = 0.063), which may be due to the limited number in the study sample. A larger sample size will be needed in future investigations. We also found that age, *MGMT* methylation, *EGFRvIII*, *p53*, *NF1*, *AxL*, *p-AxL*, and *H3Lys27* expression were associated with *CKAP2L* upregulation, and Cox regression analyses showed that older age, *IDH1*, loss of *ATRX*, and positive expression of *AxL*, *p-AxL*, *NUR77*, *H3Lys27,* or *PDGFRA* were the independent prognostic factors. We anticipate that this information will be useful to future investigations into the pathogenesis and progression of glioma. Additionally, a few unexpected results in the multivariate survival analysis, including MGMT, EGFR, EGFR vIII, and H3K27M are not statistically significant in our data. These factors are usually described as decisive prognostic markers for gliomas [10,45,46,47]. There were a few possible reasons for this. Firstly, the data shown here were the quantification of protein expression. However, the recommended analyzed tools for these genetic alterations are based on polymerase chain reaction (PCR) or fluorescence in situ hybridization (FISH). The protein expression might not be relevant to the gene expression. Take MGMT as an example; the scientist noticed the *MGMT* protein expression is poorly correlated with MGMT promoter methylation [46]. Additionally, in the *EGFR* and *EGFR vIII*, it was also found there were around 10% of cases with EGFR protein expression lacking real *EGFR* gene amplification [47,48]. Hence, precise and more proper methods might be needed for these genes’ tests. Secondly, another possible reason is that the analyzed cases were relatively small and might cause the expression of H3K27M to not reach significance. Lastly, in the recent update, the poor prognostic H3K27-mutant glioma needs to fulfill four requirements, including diffuse growth pattern located in the midline, glioma, and the evidence of H3K27M-mutant, which cannot merely rely on the staining of H3K27M [45].

It is now recognized that miRNAs play critical roles in GBM [16,17,18] and that their differential expression between malignant and normal tissue may crucially impact prognosis and potential treatments. In an effort to identify potential miRNA targets, we analyzed a miRNA array and identified miR-4496 to be the most significantly increased after *CKAP2L* knockdown (Table 4). Indeed, it appears that treating cells with exogenous miR-4496 mimics the effects of *CKAP2L* knockdown, and the effects of *CKAP2L* knockdown can be blocked by miR-4496 inhibition. However, the miRNA regulation and mechanisms are incredibly complicated [16]. Its small length could silence several mRNA targets, and the mRNA could also be regulated by more than one miRNA [16]. From our data, miR-4496 and the miR-642b-3p, and miR-4740 were revealed to be upregulated when suppressing CKAP2L. Each of them could play a potential role in the CKAP2L. Hence, these results implied that miR-4496 is a potential regulator of CKAP2L activity but needs further validation for clinical application. To date, there is no mechanism proposed to explain how CKAP2L mediates the regulation of miR-4496 and how this miRNA mediates the CKAP2L phenotype. In one study, the author noticed the low expression of decreased self-renewal ability in the colon cancer-initiating cells (CICs) through reregulated β-catenin expression via E2F1 and miR-4496 overexpression, which correlated with better prognosis [49]. In another study, they found that miR-4496 could attenuate the in vitro self-renewal and tumor-initiating capacity of CagA-expressing CICs by targeting β-catenin expression [50]. More recently, the androgen receptor could also suppress prostate cancer cell invasion through changing the miR-4496/β-catenin signals [51]. These reports all addressed the proposal that miR-4496 has a role in cancer suppression by acting on β-catenin levels and improving the outcome. Hence, it will be worthwhile to focus here and further clarify miR-4496/β-catenin signals with CKAP2L for the glioma in future work.

Our findings indicate that *CKAP2L* is an important prognostic marker in glioma. Reduction of its expression may lead to mitotic spindle dysfunction, triggering cell cycle arrest at the G2/M checkpoint, thereby suppressing cell proliferation. Moreover, by manipulating miR-4496, one may be able to suppress the oncogenic effects of *CKAP2L*, which may be a potentially useful addition to the treatments for glioma.

## 4. Materials and Methods

### 4.1. Analysis of Data from the TCGA and CGGA Databases

To understand the role of *CKAP2L* mRNA expression in glioma, we extracted the data from The Cancer Genome Atlas (TCGA; UCSC Xena) and the Chinese Glioma Genome Atlas (CGGA). The samples obtained from UCSC Xena, https://xenabrowser.net/ (accessed on 2 June 2020), contained 701 cases, including 5 normal samples and 258 grade II, 271 grade III, and 167 grade IV glioma cases. Additionally, another 325 glioma samples obtained from the CGGA database, http://www.cgga.org.cn/ (accessed on 2 June 2020), included 103 grade II, 79 grade III, and 139 grade IV cases, as well as 4 cases without grading details. Statistical comparisons of *CKAP2L* mRNA expression among tumor grades were made using multiple t-tests and one-way ANOVA. These cases were also divided into high and low *CKAP2L* expression groups around the median *CKAP2L* expression level Kaplan–Meier plot that were then constructed to compare overall survival between the high and low *CKAP2L* expression groups. Furthermore, this was done to understand the regression between CKAP2L expression versus two proliferation markers, MKI67 and MCM6. We also downloaded these datasets from The Cancer Genome Atlas (TCGA; UCSC Xena) for further investigation.

### 4.2. Gene Set Enrichment Analysis

Gene set enrichment analysis (GSEA, https://www.gsea-msigdb.org/gsea/index.jsp, accessed on 10 June 2020) was conducted by comparing the high- and low *CKAP2L* expression groups in TCGA (>5.526, *n* = 351 vs. ≤5.526, *n* = 350) and the CGGA (>1.465, *n* = 162 vs. ≤1.465, *n* = 163) datasets. Data were analyzed against the hallmark gene sets v7.1 available in the Molecular Signatures Database. For all other parameters, the default settings were used.

### 4.3. Tissue Microarray Slide Preparation, Immunohistochemistry, and Scoring

The array of brain tumor tissue, clinical information, histology diagnosis, and pathology grade (GL2083b) was purchased from GenDiscovey Biotechnology Inc., https://www.biomax.us/ (accessed on 5 April 2020), for immunohistochemistry (IHC). The commercial Tissue Array contained 132 cases of brain astrocytoma, 31 brain glioblastoma, and 9 brain oligodendroglioma, with 8 each of adjacent normal brain tissue. The core measured 1 mm in diameter with 5 µm tissue thickness. Immunohistochemical staining was performed using an automated immunostainer (Ventana Benchmark^®^XT), which ensured consistent results. Before staining, antigen retrieval was performed by heating the samples at 125 °C for 30 min in sodium citrate buffer (0.01 M sodium citrate, pH 6.2) using a pressure cooker. We switched off the cooker when it reached full pressure, and then waited till the pressure was released. Next, we took the retrieval slides and washed 3 times for 5 min each in phosphate-buffered saline (PBS). Thereafter, the slides were uploaded to the autostainer following the manufacturer’s protocol, and the selected antibodies were evaluated by checking their binding to both positive and negative controls. Following optimization of the retrieval and the antibody concentration, robust staining of the positive, but not the negative, control was observed. The test slides were then examined and scored using a semi-quantitative scoring system, same as previous publications [52,53], that took into consideration staining intensity and percentage of cells stained. The staining intensity of tumor components was scored from 0 to 3 as follows: 0 = negative, 1 = weakly positive, 2 = moderately positive, 3 = strongly positive. In addition, the percentage of tumor cells stained was estimated from 0% to 100%. Later, the percentage of cells at each intensity was multiplied by the corresponding intensity to generate an immunostaining score that ranged from 0 to 300. Finally, the immunostaining score of each sample was used for statistical analyses. The antibodies list showed as (Appendix A).

### 4.4. Statistical Analysis of the Association between CKAP2L Expression and Other Factors

To determine whether CKAP2L IHC scores correlate with tumor grades, we used Pearson’s correlation to evaluate the relation between CKAP2L expression and other possible independent factors. We also performed univariate and multivariate analyses and Cox proportional hazards regression to assess the association between factors significantly associated with CKAP2L and overall survival. The linear regression was used to evaluate the association between CKAP2L versus MKI67 and MCM6. Values of *p* < 0.05 were considered significant.

### 4.5. Human Glioma Cell Lines and Lysate Preparation

The U87MG, U118MG, and LNZ308 human glioma cell lines were maintained in Dulbecco’s modified Eagle’s medium (DMEM) supplemented with 10% fetal bovine serum (FBS), 100 U/mL penicillin, and 100 mg/mL streptomycin. Cultures were kept in an incubator under 5% CO_2_ at 37 °C. For Western blot analysis of CKAP2L expression, the cells were lysed in lysis buffer, which contained 100 mM Tris-HCl, 150 mM NaCl, 0.1% SDS, and 1% Triton X-100. GAPDH, Actin, and β-actin served as an internal control. Normal brain cell lysate purchased from Biocompare (MBS537208, San Francisco, CA, USA) also served as a control.

### 4.6. Transfection of siCKAP2L and siControl into Glioma Cell Lines and Cell Proliferation Assays

*CKAP2L* siRNA (si*CKAP2L*) and siControl were purchased from Dharmacon (Lafayette, CO, USA, product ID, M-018844-00-0005). Using DharmaFECT 1 Transfection Reagent, U87MG and U118MG cells grown in 12-well plates were transfected with 25 and 37.5 nM si*CKAP2L* or siControl (Dharmacon, USA). Cells from each group were then seeded into 12-well plates at a density of 5 × 10^4^ cells/well and incubated overnight at 37 °C. To collect the cells, 200 µL 10% FBS DMEM and 100 µL of 0.05% trypsin were added to each well. Cell proliferation was tested 24, 48, and 72 h after transfection. After preparing a mixture of 10 µL of each cell line and 10 µL trypan blue, cells excluding the dye were counted using a TC20™ Automated Cell Counter (Bio-Rad, Hercules, CA, USA). Cell proliferation assays were replicated three times with each cell line. Finally, Western blot analysis was performed to detect cell cycle checkpoint (cyclin B1, cyclin D1, p-cdc2, and cdc2) expression in the cells following si*CKAP2L* and siControl transfection.

### 4.7. RNA Isolation and Real-Time Reverse Transcription-PCR

Total RNA was extracted using TRIzol™ Reagent (Thermo Fisher Scientific, Waltham, WA, USA) and reverse transcribed to single-stranded cDNA with Tetro™ Reverse Transcriptase (Bioline, Taunton, MA, USA). For qRT-PCR, the PCR reactions were carried out using a StepOne™ Real-Time PCR System (Thermo Fisher Scientific, USA) with Fast Plus EvaGreen qPCR Master Mix (Biotium, Fremont, CA, USA). Thermocycling was performed with 0.25 µM each primer (PrimerBank) and 2.5 µL of the cDNA diluent. The PCR protocol entailed denaturation for 2 min at 95 °C followed by 40 cycles of a touch-down PCR protocol (5 s at 95 °C and 30 s annealing at 60 °C). The primers used were as follows: for *CKAP2L*, 5′-GAGCCAAAACACCAAGCCTTA-3′ (forward) and 5′-GGAGTTTAATGCTGATGGACCTT-3′ (reverse); for GAPDH, 5′-GCACCGTCAAGGCTGAGAAC-3′ (forward) and 5′-ATGGTGGTGAAGACGCCAGT-3′ (reverse).

### 4.8. Western Blot Analysis

The glioma cell lines (including U87MG, U118MG, and LNZ308) were washed three times with PBS and lysed in RIPA buffer (100 mM Tris-HCl in pH 8.0, 0.1% SDS, 150 mM NaCl, and 1% Triton 100). The protein lysates (20–40 µg, depending on the concentration) were separated by 10% SDS-PAGE and analyzed by immunoblotting with antibodies against polyclonal rabbit anti-human CKAP2L (HPA039407, Sigma–Aldrich, St. Louis, MO, USA) and anti-GAPDH (sc-47724, Santa Cruz, CA, USA) antibodies and monoclonal mouse anti-ACTN (sc-17829, Santa Cruz, USA). After repeating the experiments, the quantification was performed using ImageJ (National Institutes of Health, Bethesda, MD, USA).

### 4.9. Cell Cycle Assessment

U87MG and U118MG cells were transfected with 25 nM si*CKAP2L* for 72 h, harvested, fixed with 70% ethanol, briefly washed with PBS/1% FBS, and then incubated with 10 mg/mL RNAse A and 50 mg/mL propidium iodide in PBS plus 1% Tween 20 for 30 min at 37 °C in the dark. Flow cytometric analysis was performed using FACS Calibur flow cytometer (BD Biosciences, Franklin Lakes, NJ, USA). The cell fractions in the respective cell cycle phases were calculated using Cell Quest Pro software (BD Biosciences, USA).

### 4.10. Wound Healing Assays

U87MG glioma cells were plated in 12-well plates supplemented with mitomycin C at a density of 5 × 10^4^ cells/well and incubated overnight, after which they were transfected with si*CKAP2L* or siControl as described above. After incubation for an additional 48 h, a scratch wound was made in the cultures using a 200-µL pipet tip. The cells were then washed with PBS, and the scratched area was photographed under a microscope (0 h). One microliter of DMEM supplemented with 10% FBS was then added to each well, and after incubation for 16 h at 37 °C under 5% CO_2_, the medium was removed and the scratched area was photographed again.

### 4.11. Cell Invasion Assays

Cells were plated at a density of 5 × 10^5^ cells/dish in 10-cm dishes and incubated for 24 h, after which they were transfected with si*CKAP2L* or siControl and incubated for an additional 48 h. The transfected cells were then taken up, suspended in serum-free DMEM, and aliquots containing 2 × 10^5^ cells in 500 µL of DMEM were seeded into the upper chambers of Transwell plates, and 750 µL of DMEM supplemented with 10% FBS medium was added to the lower chambers. After incubation for 16 h, the membranes between the upper and lower chambers were fixed with formalin, stained with crystal violet, and photographed under a microscope.

### 4.12. miRNA Screening and Experiments

After transfecting U87MG and U118MG cells with si*CKAP2L* or siControl, total RNA was extracted from each sample and its quality confirmed spectrophotometrically (OD260/280 = 1.8–2, OD260/230 = 2–2.5, and RIN = 10). Samples of total RNA from the si*CKAP2L* or siControl transfectants were then screened using a Human miRNA OneArray^®^, https://www.phalanxbiotech.com/onearray-mirna-microarrays/ (accessed 13 April 2020), for analysis of differentially expressed genes encoding miRNAs. Genes were considered differentially expressed when there was a log2|Fold change| ≥ 0.585 and *p* < 0.05. In addition, miR-4496 was purchased from Dharmacon (product ID, C-302100-00-0005), after which U87MG and U118MG cells were transfected with 10 nM or 50 nM miR-4496 using DharmaFECT 1 Transfection Reagent (Dharmacon, USA). miR-4496 levels were then measured 24, 48, and 72 h after transfection. To assess the effect of miRNA inhibitor, U87MG and U118MG glioma cells were transfected with 5, 10, or 15 nM miR-4496 inhibitor (Dharmacon, USA), and cell numbers were counted 48 h later.

## Figures and Tables

**Figure 1 ijms-22-00197-f001:**
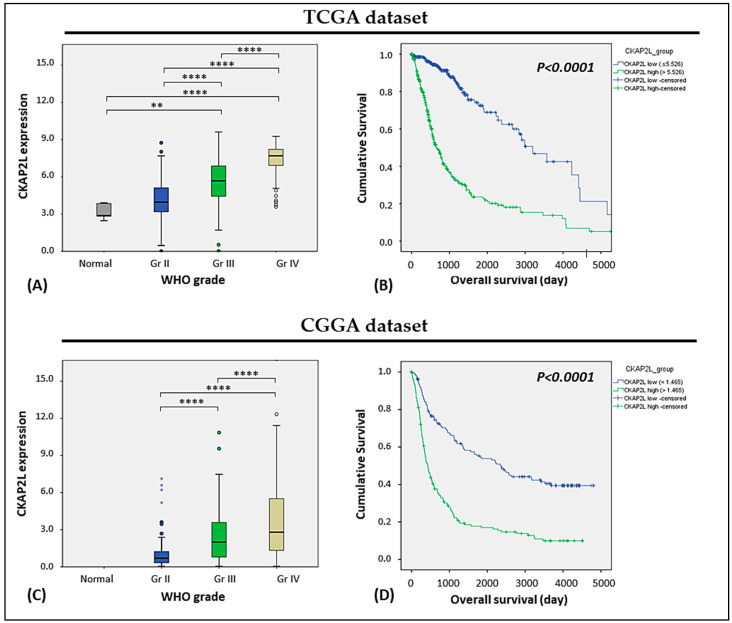
Elevated Cytoskeleton-associated protein 2-like (*CKAP2L)* mRNA expression was associated with higher tumor grade and poor prognosis in glioma. (**A**) Box plot showing that *CKAP2L* gene expression correlates with World Health Organization (WHO) tumor grade in a TCGA dataset. (**B**) Kaplan–Meier curves showing that higher *CKAP2L* mRNA expression was significantly associated with poorer prognosis in a TCGA dataset *p* < 0.00001. (**C**,**D**) Similar results were obtained with the CGGA dataset. ** *p* ≤ 0.01, **** *p* ≤ 0.0001; Kaplan–Meier curves were compared using the log-rank test.

**Figure 2 ijms-22-00197-f002:**
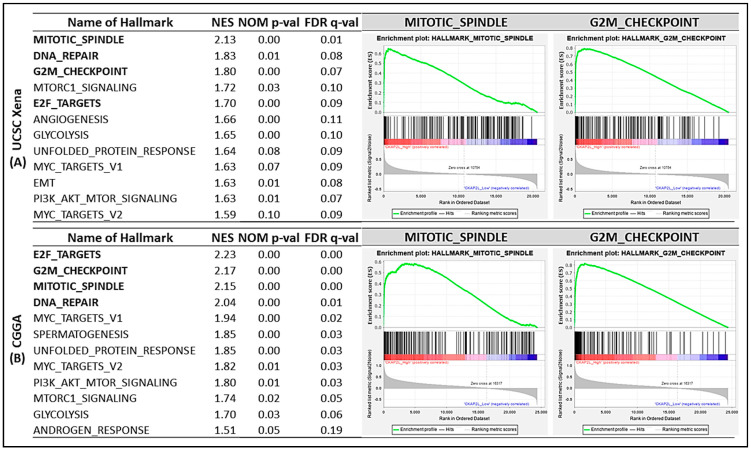
Results of gene set enrichment analysis (GSEA) in high- and low-*CKAP2L* groups in TCGA and the CGGA datasets. (**A**) In TCGA, the top phenotype in the high *CKAP2L* expression group was MITOTIC_SPINDLE, followed by DNA_REPAIR, G2M_CHECKPOINT, MTORC1_SIGNALING, and E2F_TARGETS. (**B**) Similar results were obtained with the CGGA. Note that four of the top five phenotypes were the same in both datasets.

**Figure 3 ijms-22-00197-f003:**
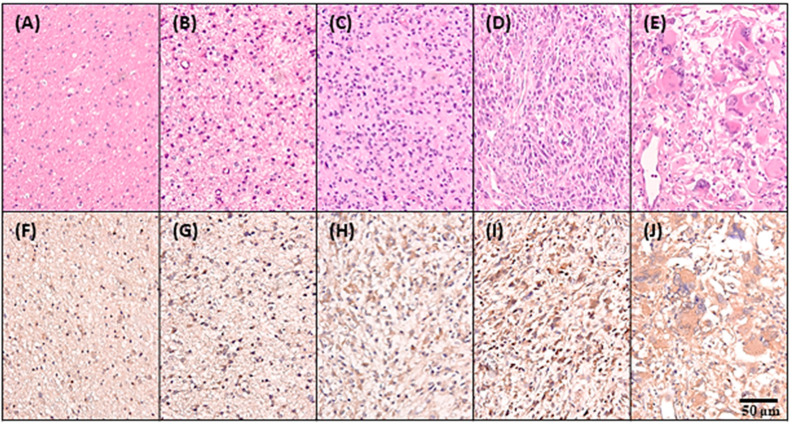
Hematoxylin/eosin and immunohistochemical staining of CKAP2L in gliomas and normal brain. Panels on the top (**A**–**E**) show H&E-stained sections, while the panels on the bottom (**F**–**J**) show immunohistochemical staining for CKAP2L. (**A**,**F**) normal brain tissue, (**B**,**G**) grade I, (**C**,**H**) grade II, (**D**,**I**) grade III, and (**E**,**J**) Grade IV. Note that CKAP2L staining is greater at higher tumor grades. Scale bar is 50 μm.

**Figure 4 ijms-22-00197-f004:**
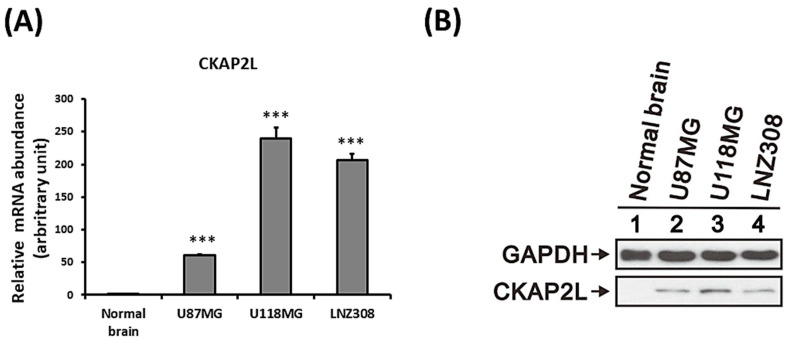
*CKAP2L* expression is enhanced in glioma cell lines. (**A**) Quantitative RT-PCR analysis of *CKAP2L* expression in normal brain tissue and the U87MG, U118MG, and LNZ308 glioma cell lines. *** *p* < 0.001. (**B**) Western blot analysis of CKAP2L expression in lysates of normal brain and U87MG, U118MG, and LNZ308 glioma cells.

**Figure 5 ijms-22-00197-f005:**
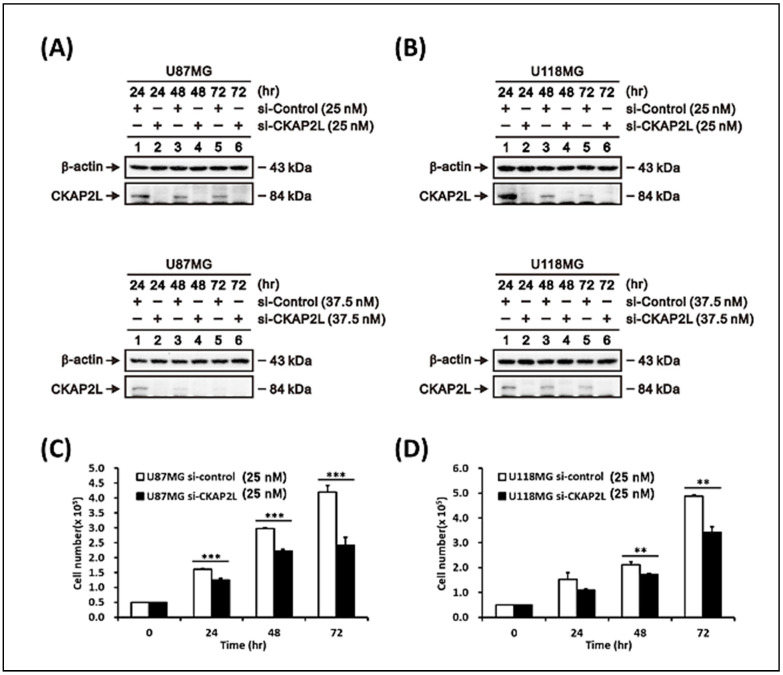
*CKAP2L* knockdown suppresses U87MG and U118MG cell growth. Transfection with si*CKAP2L* efficiently suppressed *CKAP2L* expression in (**A**) U87MG and (**B**) U118MG glioma cells with both 25 nM and 37.5 nM si*CKAP2L* or siControl treatment. (**C**,**D**) Glioma cell growth in U87MG and U118MG cells was expressing with 25 nM si*CKAP2L* or siControl at 0, 24, 48, and 72 h time points. ** *p* < 0.01; *** *p* < 0.001.

**Figure 6 ijms-22-00197-f006:**
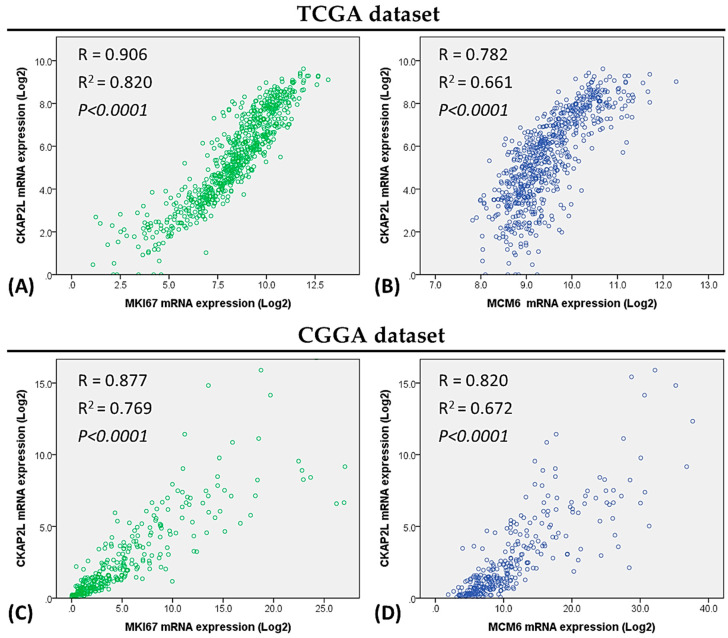
High *CKAP2L* expression positively correlates with proliferation markers, *MKI67* and *MCM6*. (**A**,**B**) Scatter plot showing that *CKAP2L* gene expression correlates with *MKI67* (R = 0.906, R^2^ = 0.82, *p* < 0.0001) and *MCM6* (R= 0.782, R^2^ = 0.661, *p* < 0.0001) in a TCGA dataset. (**C**,**D**) Similar results were obtained with the CGGA dataset (R = 0.877 and 0.820; R^2^ = 0.769 and 0.672, respectively; *p* < 0.0001).

**Figure 7 ijms-22-00197-f007:**
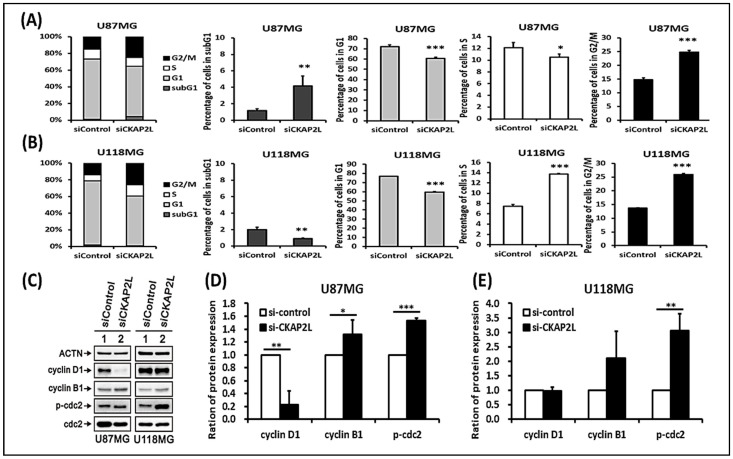
*CKAP2L* induces cell cycle G2/M arrest. (**A**,**B**) Cell cycle profiles in U87MG and U118MG cells. Note that *CKAP2L* knockdown increases cell fraction in G2/M phase and decreases the fraction in G1 phase. (**C**) Representative Western blot showing the effect of *CKAP2L* knockdown on expression of cell cycle regulators in U87MG and U118MG glioma cells. (**D**,**E**) Quantitative analysis of the Western blot data showing increased expression of cyclin B1 and p-cdc2 after *CKAP2L* knockdown. * *p* < 0.05; ** *p* < 0.01; *** *p* < 0.001.

**Figure 8 ijms-22-00197-f008:**
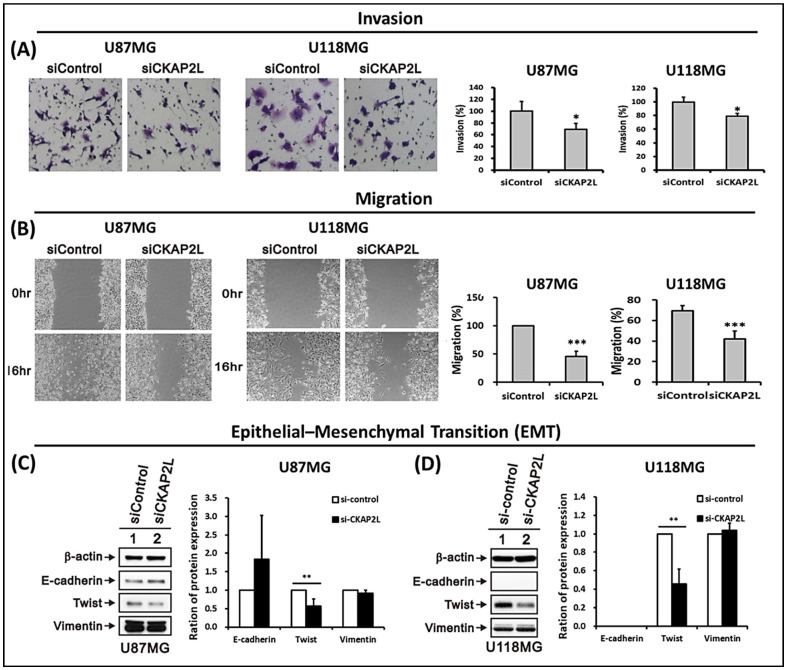
*CKAP2L* knockdown reduces invasion, migration, and epithelial-mesenchymal by U87 and U118 glioma cells. (**A**) On the left are representative photomicrographs of the underside of transwell filters showing that fewer si*CKAP2L*-expressing cells crossed the membrane from the upper chamber. Relative levels of invading cells are quantitated on the right. (**B**) On the left are photomicrographs of scratch wounds showing that *CKAP2L* knockdown reduces migration into the gap. Relative levels of migrated cells are quantitated on the right. (**C**,**D**) Representative Western blot showing the effect of *CKAP2L* knockdown on the expression of the indicated epithelial-mesenchymal transition (EMT) markers. Quantitative analysis of the Western blot data showing a trend toward reduction in Twist. * *p* < 0.05; ** *p* < 0.01; *** *p* < 0.001.

**Figure 9 ijms-22-00197-f009:**
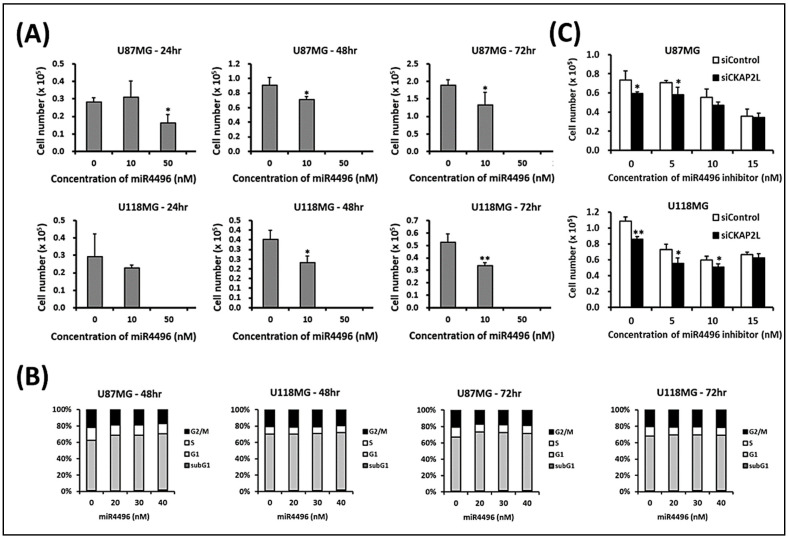
Exogenous application of miR-4496 to U87MG and U118MG cells mimics the effect of *CKAP2L* knockdown on cell proliferation. (**A**) Bar graphs showing the concentration-dependent inhibitory effect of miR-4496 on cell proliferation. (**B**) Cell cycle analyses of U87MG and U118MG at 48 and 72 h in the gradient concentration of miR-4496 from 0 to 40 nM. (**C**) miR-4496 inhibitor concentration dependently suppressed the anti-proliferative effect of si*CKAP2L*, eliminating the difference between si*CKAP2L*-expressing and siControl-expressing cells. * *p* < 0.05; ** *p* < 0.01.

**Table 1 ijms-22-00197-t001:** CKAP2L immunostain scores in gliomas.

	Average Intensity	Average % Tumor	Average Score	Correlation *
Non-neoplastic brain tissue	0.67	10	10	
**Classification of gliomas**				
Pilocytic astrocytoma	1	5	5	No correlation (*p* = 0.057)
Diffuse astrocytoma, IDH1-mutant	1	20	20	
Diffuse astrocytoma, IDH1-wildtype	1	26.82	26.82	
Anaplastic astrocytoma, IDH1-mutant	1	23.33	23.33	
Anaplastic astrocytoma, IDH1-wildtype	1	40	40	
Glioblastoma, IDH1-mutant	1	42.5	42.5	
Glioblastoma, IDH1-wildtype	1	36.72	39.06	
Oligodendroglioma, NOS	0.75	13.75	13.75	No correlation (*p* = 0.960)
Anaplastic oligodendroglioma, NOS	0.67	13.33	13.33	
Diffuse midline glioma, H3 K27M-mutant	1.08	39.58	50.42	
**WHO grades of gliomas**				
WHO grade I	1	5	5	Positive correlation (*p* = 0.005 *)
WHO grade II	0.94	23.13	23.13	
WHO grade III	0.86	25	25	
WHO grade IV	1.02	38.10	42.20	

* Differences were analyzed using the Pearson correlation method.

**Table 2 ijms-22-00197-t002:** Univariate and multivariate analysis of risk factors associated with CKAP2L expression.

		Univariate Analysis		Multivariate Analysis	
Variable	Total	OR (95% CI)	*p* Value	OR (95% CI)	*p* Value
Age					
<50	32	1			
≥50	39	1.72 (1.48–2.28)	0.010 *	1.72 (1.48–2.26)	0.042 *
MGMT					
Unmethylated	31	1			
Methylated	40	1.07 (1.03–1.16)	0.714	1.08 (1.04–1.18)	0.008 *
EGFRvIII					
Negative	54	1			
Positive	17	1.42 (1.29–1.50)	0.089	1.49 (1.39–1.55)	0.012 *
P53					
Negative	35	1			
overexpression	36	1.39 (1.31–1.55)	0.081	1.44 (1.32–1.68)	0.007 *
NF1					
Negative	37	1			
Positive	34	1.39 (1.32–1.53)	0.081	1.40 (1.32–1.58)	0.013 *
AxL					
Negative	25	1			
Positive	46	1.23 (1.12–1.48)	0.297	1.27 (1.12–1.62)	< 0.001 *
p-AxL					
Negative	18	1			
Positive	53	1.16 (0.98–1.63)	0.512	1.12 (0.96–1.55)	0.017 *
H3Lys27					
Negative	9	1			
Positive	62	2.39 (0.38–1.36)	0.020 *	2.47 (1.23–1.31)	0.010 *

* indicates the statistical significance, *p* < 0.05.

**Table 3 ijms-22-00197-t003:** Multivariate analysis for overall survival in gliomas.

	Multivariate Analysis		
Variable	Hazard Ratio	95% Confidence Interval	*p* Value
Sex (male/female)	1.69	0.70–4.06	0.240
Age (<50/≥50)	6.05	2.36–15.52	<0.001 *
CKAP2L (≤20/>20)	0.43	0.17–1.05	0.063
IDH1 R132H (Negative/Positive)	2.87	1.05–2.87	0.040 *
ATRX (Preserve/Loss)	0.36	0.17–0.79	0.010 *
H3K27M (Negative/Positive)	0.60	0.19–1.96	0.398
MGMT (Unmethylated/ Methylated)	0.46	0.19–1.08	0.073
EGFR (Negative/Positive)	5.99	0.96–37.56	0.056
EGFRvIII (Negative/Positive)	1.10	0.44–2.77	0.845
P53 (Negative/Overexpression)	1.89	0.85–4.21	0.119
Neurofilament (Negative/Positive)	0.28	0.09–0.81	0.019
NF1 (Negative/Positive)	1.69	0.73–3.89	0.222
AxL (Negative/Positive)	2.75	1.17–6.51	0.021 *
p-AxL (Negative/Positive)	4.49	1.53–13.16	0.006 *
NUR77 (Negative/Positive)	0.35	0.15–0.83	0.018 *
H3Lys27 (Negative/Positive)	0.33	0.11–0.99	0.049 *
PDGFRA (Negative/Positive)	9.73	1.74–54.48	<0.001 *

* indicates the statistical significance, *p* < 0.05.

**Table 4 ijms-22-00197-t004:** *CKAP2L* knockdown induces upregulation of miR-4496, miR-642b-3p, and miR-4740 in U87 and U118 glioma cells.

U87	Normalized Intensity		Coefficient of Variation		log2FC *	*p*-Value
Name of miR	U87 with siControl	U87 with si*CKAP2L*	U87 with siControl	U87 with si*CKAP2L*	(si*CKAP2L*/siControl)	(si*CKAP2L*/siControl)
hsa-miR-642a-3p	467.39	3253.615	0.08381377	0.23918642	2.79934478	0.036951922
hsa-miR-4496	90.5	405	0.16408003	0.21300501	2.16193221	0.036620139
hsa-miR-642b-3p	144.875	391.245	0.18913124	0.1822687	1.43326365	0.04486455
hsa-miR-132-3p	56	101.75	0.22728432	0.02432308	0.86153006	0.03789425
hsa-miR-4740-3p	33.5	55.5	0.02110767	0.01274066	0.72832668	0.00103146
hsa-miR-4712-3p	65	106	0.13054279	0.05336655	0.70555264	0.02956881
hsa-miR-10a-3p	48	76.5	0.02946278	0.06470258	0.67242534	0.015924106
hsa-miR-491-3p	52.5	81	0.0404061	0.03491885	0.62560449	0.007606987
hsa-miR-4723-5p	4147.415	6316.36	0.08147367	0.01306662	0.60688099	0.01261677
**U118**	**Normalized Intensity**		**Coefficient of Variation**		**log2FC ***	***p*-Value**
**Name of miR**	**U118 with** **siControl**	**U118 with** **si*CKAP2L***	**U118 with** **siControl**	**U118 with** **si*CKAP2L***	**(si*CKAP2L*/siControl)**	**(si*CKAP2L*/siControl)**
hsa-miR-668-3p	475.865	766.455	0.06281068	0.01006522	0.68764874	0.005594934
hsa-miR-4740-3p	33.5	63	0.02110767	0.0673435	0.91119073	0.010462609
hsa-miR-642b-3p	116.165	390.805	0.087228	0.02241797	1.75027347	0.001187312
hsa-miR-4496	89.5	469.16	0.05530444	0.04473299	2.39012043	0.001608935

* log2FC, FC indicates “Folds change”.

## Data Availability

The data presented in this study are available in the supplementary material.

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
