# Peer review of "CKAP2L Knockdown Exerts Antitumor Effects by Increasing miR-4496 in Glioblastoma Cell Lines"

_ijms, 2020, doi:10.3390/ijms22010197_

Round 1
Reviewer 1 Report
The authors really improved the quality of the manuscript in this new version.
They completely addressed most of my comments. However, I still believe that some of their conclusion are based only on speculations and hypothesis and could be improved with few additional experiments.
In particular:
- The data about the possibility that the impact of CKAP2L alteration in cancer is only mediated by effect on proliferation, and that its downregulation only affect this aspect, are still not convicing at all. Although the authors reported additional information that could support this hypothesis (Supplementary_4), I still recommend to add EdU proliferation assay or colony formation assay for testing cell proliferation in a direct manner and a TUNEL assay to analyse cell survival, expecially in the experiment with the miR-4469 (in parallel with siCKAP2L experiments) reported in figure 8.
- An interesting aspect is the possibility that CKAP2L (and miR-4469) are involved in the formation of multipolar spindles, as suggested by the GSEA data (figure 2) and discussed by the authors. Additional experiments by immunofluorescent analysis using an a-tubulin antibody in siCKAP2L (and siControl), overexpression and inhibition of miR-4496 (alone and in siCKAP2L treated cells) cell lines, can easily stain the microtubules of mitotic spindle, and analyze the percentage of mitotic cells exhibiting multipolar spindles. This experiments could allow the validation of this hypothesis, and increase the quality of the manuscript and the reported findings
Author Response
Thanks for these fantastic comments, which do help to improve our manuscript. Please see the attachment for the details.

Reviewer 2 Report
The authors have addressed the concerns adequately that were raised with the initial version of the manuscript.
Author Response
Many thanks for the reviewer's fantastic comments and improved the quality of our work on "CKAP2L knockdown exerts antitumor effects by increasing miR-4496 in glioblastoma cell lines". The reviewer's suggestions have been very precious for revising our manuscript. Thanks again.
Reviewer 3 Report
All corrections have been adressed
Author Response

(The authors gave the same response as above.)

Round 2
Reviewer 1 Report
The authors now moved the original Supplementary_4 into the main manuscript and now it became Figure 6. This make the manuscript clearer.
However, without some validation experiments I still believe that the conlcusions are overstated.
In particular I think that it is missing the connection between what they show in the GSEA analysis and in the siCKAP2L experiments with what they show in the miR-4496 experiments (expecially I would like to underline again that they show a "similar" effect on proliferation when they inhibit or overexpress it). If they do not want to investigate further on miR-4496 function I strongly recommend to modify the conclusions:
In line 253 they should mention a possible effect mediated by the other upregulated miRNAs and that the "suppressing CKAP2L decreased cell proliferation in part by increasing levels of miR-4496", since as they also mention in their response to reviewer "From our data, miR-4496 and the miR-642b-3p, and miR-4740 revealed up-regulated when suppressing CKAP2L. Each of them could play a potential role in the CKAP2L."
Accordingly, also the sentence in the discussion in line 381 "These results suggest that miR-4496 is a vital regulator of CKAP2L activity." should be modified and discussed.
Finally, since they do not really demonstrate the presence of multipolar spindles the sentence in line 340 has to be modified accondingly:
"we therefore hypothesized that the arrested cells were blocked at the G2/M checkpoint due to the possible presence of multipolar spindles [37] that may be caused by suppression of CKAP2L expression."
Author Response
Many thanks for the reviewer's fantastic comments and improved the quality of our work. As reviewers' advice, we have responded to them individually and highlighted the changes we made. Please see the attachment. Thanks again.

This manuscript is a resubmission of an earlier submission. The following is a list of the peer review reports and author responses from that submission.
Round 1
Reviewer 1 Report
This is a well-written and performed study that warrants publication. However some areas need to be addressed/clarified :
- Can the authors explain why they used commercial cell lines that have been in use in the literature for many years. Did they consider using patient derived cell lines, since they are affiliated with hospitals ?
2. Lines 38-40, they mention that diagnoses were based on cellular morphology etc prior to 2016 without stating that this was the WHO Classification for Brain Tumours.
3. The kaplan meier curves in figure 1 combine the 3 grades of glioma. Is there a survival difference based on CKAP2L expression when survival is examined within each separate grade. Can these be included as supplementary figures ?
4. Lines 110-111 : After eliminated cases that lacked adequate specimens.
Can the authors please explain this comment as all of the tissue cores in a commercial TMA should be viable for use
5. Line 148 - western blots.
Can the authors state the amount of protein that they routinely loaded into their western blots
6. Section 2.7 - CKAP2L knockdown inhibits invasiveness and epithelial-mesenchymal transition (EMT) in glioma cells
Why were the invasion and migration assays only carried out with the U87MG cell line, when other experiments were performed with at least one of the two remaining cell lines ?
7. Line 293 - samples were heated at 125C for 30 min.
Can the authors elaborate on how the antigen retrieval was performed ? What equipment was utilized for this ?
8. Lines 363-365
Did the authors pre-treat their cells with mitomycin C to stop proliferation during the migration assay, as the differences observed may be due to a difference in cell proliferation as demonstrated in figure 5
Reviewer 2 Report
CKAP2L knockdown exerts antitumor effects by increasing miR-4496 in glioblastoma cell lines
Review
This article presents a very large amount of data. The experiments are described in detail, and the message is clear. A good part of the experiments are on the functional role of CKAP2L on the proliferation, the migration and the invasion of the cell lines.
I suggest that authors make some modifications that I have mentioned in the list below.
Introduction
Line 35 CNS malignancies are not among the most frequently occurring cancers (not even among the 10 most frequent in adults), in children it is most frequent after hematological malignancies in infants. It is not very relevant for the rest of the paper.
Line 38, the official name is glioblastoma, not astrocytoma grade IV, as not so called glioblastoma. Line 38-40 is irrelevant.
Line 43: played a critical role in the prognosis. It is only 1p19q codeletion that has a definitive predictive value on treatment choice.
Line 55. CKAP2L gene. Probably the authors should make a notice, because at other place in the text they refer to CKAP2 or CKAP2L also as protein.
Results
Line 70, no difference between normal brain and grade II tumors, …likely reflects the small number of normal brain samples. Do we know where the normal brain is coming from? Is it from contralateral brain, från epilepsy surgery, from injury or post mortem ?
Line 79, and referral to figure 1 B,D. there is significant difference in median cut-off in TCGA high CKAP2L (> 5.526) and median cut-off I CGGA (> 1.465. In the figures, it seems that the CGGA have better overall survival in both low and high CKAP2L. Furthermore, the scale in the Kaplan Meier graphs are not the same for time, it would help in comparison. Could the authors give an explanation of these differences (in the discussion)? In section 4.1 it appears that the TCGA contained many more cases of grade II and III tumors compared to glioblastoma (grade IV), in the CGGA there were more grade IV than II and III, but the survival was better in the latter anyway.
Figure 2 is practically impossible to read (too small)
Figure 3 shows the correlation of immunostaining with higher grade which is expected. Since CKAP2L gene product is related to the mitotic process, would it be possible to indicate for example the amount of mitoses per high power field (which also increase with tumor grade )?
Table 1 I tried to figure out what average intensity number represents, as well as average % tumor, average score, but I could find, could the author explain a little more. To which line does “no correlation” refer to: IDH1 wt to mutant or average tumor to average score ? What is the additional information between detailed “classification of gliomas” and lower down “WHO grades of gliomas”
Table 2 is too busy. I think it more informative to only keep the statistically significant values (I suppose there are marked *, nothing in the legend), and only multivariate analysis. The only 2 univariate analysis that are significant are also in the multivariate. Is # also a sign for significant ?
In the table, H3K27M appears 2 times, once with 61 negative and 10 positive, and later with 9 negative and 62 positive, I suppose that the latter is not H3K27M but ratherH3Lys27 ?
Table 3 gives some unexpected results, because the multivariate analysis for survival is for example not statistically significant for MGMT, EGFR, EGFR vIII and H3K27M. These factors usually have been described previously as prognostic for the survival of gliomas, is there an explanation ?
Table 4: I would appreciate a legend for normalized intensity : SC ?, ST?, CV ?
Discussion
Line 230 CKAP2 is an important paralog of CKAP2L, if both are genes, but I am not sure that the authors means CKAP2 is a gene or a protein.
The discussion is relatively short, and repeating part of the results.
Line 252 and 254, I suppose that H3K27M in both lines are not correct
If CKAP2L is an independent factor for prognosis, and it follows very nicely the Hematoxylin-Eosin staining in the grade of the tumor, what would this test add for the prognosis, I believe the authors have not definitely proved that it gives something more to the prognosis. CKAP2L explain certainly how the tumor is spreading and invading, and that it may in the future become targetable, but what about its prognostic role in the day to day routine?
Material and methods
Line 359, line 367, line 369: 5x104 cells should be 5x104, same in the other lines
Reviewer 3 Report
In my opinion, the results of this study can be published in publishing house MDPI after minor revision. But for the IJMS the level of research is insufficient. At least conclusions drawn from the last part of the manuscript concerning miR-4496 are insufficiently confirmed by experiment. I could recommend this manuscript for publication in some other lower-rated journal.
Reviewer 4 Report
In this article the authors investigated the role of Cytoskeleton-associated protein 2-like (CKAP2L) in glioma. They found that CKAP2L expression correlates with tumor grade and CKAP2L knockdown inhibits glioma cell proliferation, migration, invasion, and epithelial-mesenchymal transition. Their data suggest that CKAP2L knockdown led to significant increases in miR-4496 and the effects of CKAP2L knockdown could be suppressed by miR-4496 inhibition. Although the findings are interesting and opens perspective in the use of CKAP2L as potentially useful prognostic marker in glioma, some of the data are insufficient to claim such conclusions and requires additional work before to be published.
- Statistical analysis showed a strong correlation between CKAP2L mRNA expression and prognosis in both TCGA and the CGGA datasets. Although the statistical analysis (multiple t-tests and one-way ANOVA) showed high significance, the error bars reported in figure 1A and 1C are very big. The authors should discuss this big variability.
- To further explore CKAP2L mRNA and protein expression in glioma, they performed quantitative RT-PCR and Western Blot (WB) analyses with normal brain tissue and the U87MG, U118MG, and LNZ308 human glioma cell lines. WB in figure 4B, show no expression of CKAP2L protein in normal brain tissue. I cannot see any band. In this respect, how can they normalize the protein levels in the human glioma cell lines respect to the normal brain tissue? I think that the data should be described differently. In addition, I would like to see the Ct mean values and the ΔCt of CKAP2L mRNA in normal brain tissue because, based on the protein bands, I’m expecting that also the mRNA levels are absent or very low. Again, if this is the case, data should be described differently, probably in absolute value.
- In Figure 5A and 5B the authors showed that after transfection of siCKAP2L or scramble siRNA into U87MG and U118MG glioma cells expression of CKAP2L protein was reduced. The authors claim that the reduction is substantial. Although I can see a reduction in U87MG cell line, I could not really appreciate the reduction in U118MG cell line from the representative WB showed in figure 5B. Moreover, the ratio of CKAP2L protein levels in siCKAP2L samples respect to the scramble siRNA ones, indicate a reduction of 31% in U87MG cell line and 23% in U118MG cell line. This reduction in insufficient to consider this experiment as a silencing, even more considering how much their levels are increased respect to the normal brain tissue. The authors reported, in the material and methods section, that they used 25 nM siCKAP2L for these experiments. Is it possible to increase the siCKAP2L concentration to obtain a stronger decrease in CKAP2L protein levels? These WB control experiments at which time point have been performed (24h, 48h,72h)?
- The authors evaluated cell proliferation 24, 48, and 72 h after the siRNA transfection revealing that within 24 h in U87MG cells or 48 h in 156 U118MG cells, cell proliferation was significantly lower in siCKAP2L-expressing cells than in the control group (Figure 5C and 5D). However, in Figure 5C and 5D I see reported only cell count upon transfection at different time points. Although the authors stated, in the material and method section, that they tested both cell viability (trypan blue) and proliferation I cannot see these data along the entire manuscript. Since the authors do not show any data about cell death, how they can be sure that the reduction in the number of cells is due to the reduction in proliferation and not to an increase of cell death? To really state that there is a reduction in proliferation, without increase of cell death, the authors should performed an EdU proliferation assay or colony formation assay and a TUNEL assay in siCKAP2L versus scramble siRNA in U87MG and U118MG glioma cells.
- The authors compared the cell cycle phase distributions between glioma cells expressing siCKAP2L and siControl, concluding that CKAP2L knockdown arrested the glioma cells at the G2/M phase. This conclusion is based on the assumption that CKAP2L knockdown reduced cell proliferation that, as suggested in the previous point, should be better validated. Moreover, they also showed the increased levels of cyclin B1 and p-cdc2, but in the same panel they also showed a reduction of cyclinD1 in U87MG cell line and no variation in U118MG cell line. How the authors interpret this data considering that the two cell lines behaves differently in S phase and cyclinD1 is usually dramatically decrease in S phase (1101/gad.7.5.812 , 10.1038/sj.onc.1208326, 10.1186/1747-1028-1-32)? I think this data should be discussed.
- Finally, using a miRNA array in U87MG and U118MG glioma cells expressing siCKAP2L or siControl, they identified three miRNAs that were upregulated after CKAP2L knockdown. The authors stated that miR-4496 overexpression in U87MG and U118MG glioma cells decrease cell proliferation, mimicking CKAP2L knockdown effect. However, as suggested in point 4, this data needed further experiments to exclude an effect on cell death instead of a reduction of proliferation. Moreover, they showed that overexpression of miR-4496 reduced cell number, but inhibition of this miRNA in both cell line also reduced cell number. Indeed, by looking at the white bar (siControl) of the figure 8B, it is evident that increased inhibition of miR-4496 causes cell number decrease in U87MG and U118MG glioma cells. Although it seems that miR-4496 inhibition counteracts siCKAP2L effect on cell number, a better characterization of miR-4496 (both inhibition and overexpression) effect on cell cycle and cell survival should be performed.
In the discussion the authors conclude that they hypothesized that the arrested cells were blocked at the G2/M checkpoint due to the presence of multipolar spindles caused by suppression of CKAP2L expression. This hypothesis should be further corroborated by immunofluorescent analysis of siCKAP2L and siControl cell lines using an a-tubulin antibody, that can easily stain the microtubules of mitotic spindle, and analyze the percentage of mitotic cells exhibiting multipolar spindles. The authors propose that CKAP2L function is exerted via miR-4496. However, to confirm this hypothesis the analysis of mitotic spindle should be performed in parallel with the overexpression and inhibition of miR-4496 (alone and in siCKAP2L treated cells).
No mechanism is proposed to explain how CKAP2L mediates regulation of miR-4496 and how this miRNA mediate the CKAP2L phenotype. Which are the miR-4469 predicted target? Which could be a possible mechanism? miR-4496 is reported to have a role in cancer suppression by acting on β-catenin levels, the authors should at least discuss their data considering these other findings reported in literature. (10.1016/j.bbrc.2018.08.134, 10.1002/path.4866, 10.1084/jem.20141254).
At moment, the reported experiments do not exhaustively support the author conclusion that “miR-4496 is a vital regulator of CKAP2L activity”.